# The Use of Telemedicine in Nursing Homes: A Mixed-Method Study to Identify Critical Factors When Connecting with a General Hospital

**DOI:** 10.3390/ijerph182111148

**Published:** 2021-10-23

**Authors:** Clément Cormi, Jan Chrusciel, Antoine Fayol, Michel Van Rechem, Khuloud Abou-Amsha, Matthieu Tixier, Myriam Lewkowicz, David Laplanche, Stéphane Sanchez

**Affiliations:** 1Pôle Territorial Santé Publique et Performance des Hôpitaux Champagne Sud, Centre Hospitalier de Troyes, 10000 Troyes, France; jan.chrusciel@hcs-sante.fr (J.C.); david.laplanche@hcs-sante.fr (D.L.); stephane.sanchez@hcs-sante.fr (S.S.); 2LIST3N/Tech-CICO, Troyes University of Technology, CEDEX, 10300 Troyes, France; khuloud.abou_amsha@utt.fr (K.A.-A.); matthieu.tixier@utt.fr (M.T.); myriam.lewkowicz@utt.fr (M.L.); 3Hôpital Européen Georges Pompidou, AP-HP, Université de Paris, 75015 Paris, France; antoine.fayol@aphp.fr; 4Service des Urgences, Centre Hospitalier de Troyes, 10000 Troyes, France; Michel.Vanrechem@ch-troyes.fr; 5University Committee of Resources for Research in Health (CURRS), University of Reims Champagne-Ardenne, CEDEX, 51095 Reims, France

**Keywords:** telemedicine, nursing homes, rural health, urban health, health services

## Abstract

Evaluating the use and impact of telemedicine in nursing homes is necessary to promote improvements in the quality of this practice. Even though challenges and opportunities of telemedicine are increasingly becoming well documented for geriatrics (such as improving access to healthcare, patient management, and education while reducing costs), there is still limited knowledge on how to better implement it in an inter-organizational context, especially when considering nursing homes. In this regard, this study aimed first to describe the telemedicine activity of nursing homes when cooperating with a general hospital; and then understand the behavioral differences amongst nursing homes while identifying critical factors when implementing a telemedicine project. We conducted a sequential, explanatory mixed-method study using quantitative then qualitative methods to better understand the results. Three years of teleconsultation data of twenty-six nursing homes (15 rural and 11 urban) conducting teleconsultations with a general hospital (Troyes Hospital, France) were included for the quantitative analysis, and eleven telemedicine project managers for the qualitative analysis. Between April 2018 and April 2021, 590 teleconsultations were conducted: 45% (*n* = 265) were conducted for general practice, 29% (*n* = 172) for wound care, 11% (*n* = 62) for diabetes management, 8% (*n* = 47) with gerontologist and 6% (*n* = 38) for dermatology. Rural nursing homes conducted more teleconsultations overall than urban ones (RR: 2.484; 95% CI: 1.083 to 5.518; *p* = 0.03) and included more teleconsultations for general practice (RR: 16.305; 95% CI: 3.505 to 73.523; *p* = 0.001). Our qualitative study showed that three critical factors are required for the implementation of a telemedicine project in nursing homes: (1) the motivation to perform teleconsultations (in other words, improving access to care and cooperation between professionals); (2) building a relevant telemedicine medical offer based on patients’ and treating physicians’ needs; and (3) it’s specific organization in terms of time and space. Our study showed different uses of teleconsultations according to the rural or urban localization of nursing homes and that telemedicine projects should be designed to consider this aspect. Triggered by the COVID-19 pandemic, telemedicine projects in nursing homes are increasing, and observing the three critical factors presented above could be necessary to limit the failure of such projects.

## 1. Introduction

People older than 65 account for 16.5% and 20.3% of the US population and the European population, respectively [1,2]. An aging population has increasing healthcare needs and requires more healthcare facilities [3,4]. The Administration for Community Living (ACL) estimates that 35% of US citizens older than 65 will become nursing home residents [5]. However, the availability of physicians in nursing homes is often limited notably because of less advantageous work conditions [6,7], which forces staff to hospitalize residents despite associated medical risks [8,9].

Two strategies have been used to alleviate the lack of medical resources: employing advanced nurse practitioners and relying on telemedicine to remotely connect with physicians [10,11]. Initially restricted due to technical limitations, telemedicine use has been increasing which has allowed for improved access to healthcare, patient management, and education while reducing costs (mostly associated with transportation) [12,13,14,15,16,17,18,19,20,21]. In the context of the COVID-19 pandemic, telemedicine use in nursing homes is clearly relevant in that it limits interruptions in care and reduces nursing home residents’ exposure to COVID-19 in hospitals [22].

Telemedicine makes it possible for a wide variety of actors (patients, health professionals, or families) to connect and can address a large number of situations (programmed care and follow-up, non-programmed care, and even emergencies) [23]. However, trials and pilot projects in telemedicine often struggle to become sustainable, especially when an initial project is supported by a single individual [24]. Current literature highlights some barriers associated with this, including funding, time, infrastructure, equipment, skills, and preference for a face-to-face consultation [25]. Some ethical issues have also arisen, including autonomy, beneficence, non-maleficence, justice, and professional-patient relationships [26]. Nevertheless, studies have shown to agree that telemedicine projects should not be driven by technology but rather by the users’ needs and objectives [27,28,29].

Geriatrics provides some of the strongest evidence regarding the use of telemedicine [30], however, most only consider interventions with home-based patients [31]. For nursing homes, May et al. stated that the adoption of telemedicine often is hindered by the disruption it causes in the daily work organization including additional administrative efforts and interruptions in the daily care routine [32]. Regardless, additional data are continuously needed to document the success and failures of telemedicine projects in nursing homes because of the complexity of their inter-organizational and inter-professional collaboration components, as stated in a recent scoping review [31]. Consequently, a three-year study on a synchronous video teleconsultation program conducted between a general hospital and several nursing homes in our region was conducted [33]. The objectives of the present study were first to describe the telemedicine activity of nursing homes cooperating with a general hospital (quantitative study) and understand the behavioral differences amongst nursing homes while identifying critical factors when implementing a telemedicine project (qualitative study).

## 2. Materials and Methods

A mixed-method analysis was performed using both quantitative and qualitative methods. It was conducted since it provides a good understanding of large and complex environments such as those faced in telemedicine activities [34,35]. Based on the Caffery et al. method, we used a sequential, explanatory design starting with a quantitative analysis and continuing with a qualitative study to better understand the results [36].

### 2.1. Quantitative Analysis

The quantitative study was a monocentric and retrospective analysis of teleconsultation activity involving 26 nursing homes (15 rural and 11 urban) and a general hospital (Troyes Hospital) located in the same department over the period from April 2018 to April 2021.

Quantitative variables were described as mean ± standard deviation (SD) and categorical variables as number and percentage. Multivariate analyses of the total teleconsultation activity and the specific activity for general practice were conducted using the negative binomial model. Analyses were adjusted based on the time (aggregated 1-year periods), location (urban or rural), and the number of residents in the nursing homes. A *p*-value < 0.05 was considered statistically significant. All analyses were performed using SPSS 21.0^®^ (SPSS, Chicago, IL, USA).

### 2.2. Qualitative Analysis

The qualitative study relied on semi-structured interviews with the telemedicine project managers of the concerned nursing homes. They consisted of facility directors, coordinating physicians, and/or nurses. The qualitative study aimed to explain the differences in the behaviors of nursing homes and was revealed by our quantitative study while identifying critical factors that could be considered when implementing a telemedicine project.

Each telemedicine project manager from all the nursing homes was invited to participate voluntarily and without any advantages whatsoever. Participants were first invited by email, then by phone if they did not reply to the first email. Interviews were conducted by the same investigator (CC), who is a male and has a PhD, and is a member of the Troyes Hospital Public Health Research team. Each interview included questions related to the nursing home’s overall strategy to provide for all medical needs, their current use of telemedicine, and their potential needs related to telemedicine.

The interview question guide was tested with a telemedicine project manager and did not need any modifications (Appendix A). In total, 11 interviews were conducted to reach data saturation [37]. With the oral consent of the participants, interviews were recorded and were subsequently transcribed and anonymized.

The data of the qualitative study used a thematic analysis approach with the aim of identifying and categorizing the various themes occurring across all interviews in a cross-sectional manner [35,38]. Each theme was then considered as a meaningful and independent unit of discourse. Major themes and secondary themes were identified.

Major themes are relevant topics that are spontaneously well developed by all participants. Minor themes are less well developed by the participants and are of lesser importance in their discourse. Transcripts were analyzed by the three co-authors (KAA (female), MT, and CC (male)).

The first round of analysis was performed by each of the three co-authors individually. The second round was performed jointly by the same co-authors after all the material had been collected. The second-round analysis resolved differences in interpretation. All interviews were held, recorded, transcribed, and analyzed in French. Key citations were translated for publication purposes and to illustrate qualitative results.

## 3. Results

### 3.1. Quantitative Study

In total, 590 teleconsultations were conducted during the three-year period (see Table 1). The rural nursing homes were located on average 39 ± 13 km from the hospital, in municipalities with an average of 1904 ± 958 inhabitants, and housed an average of 86 ± 32 residents. Moreover, the urban nursing homes were located an average of 3 ± 1.6 km from the hospital, eight of which were located in the same city of 61,000 inhabitants as the hospital. These had an average of 88 ± 25 residents.

During the period, seven nursing homes conducted about 80% of the teleconsultations (480 on 590), three rural nursing homes conducted 80% of the teleconsultation conducted by all the rural nursing homes (367 on 463), and five urban nursing homes did the same (105 on 127).

Among the 590 teleconsultations, 45% (*n* = 265) were conducted for general practice, 29% (*n* = 172) for wound care, 11% (*n* = 62) for diabetes management, 8% (*n* = 47) with gerontologist and 6% (*n* = 38) for dermatology (Table 2). Rural nursing homes conducted more teleconsultations overall than urban ones (RR: 2.484; 95% CI: 1.083 to 5.518; *p* = 0.03) and included more teleconsultations for general practice (RR: 16.305; 95% CI: 3.505 to 73.523; *p* = 0.001) (Table 3).

### 3.2. Qualitative Study

Out of the 26 requests, thirteen interviewees replied positively to our invitation. Five were not available due to COVID-19 management constraints, and eight did not reply. Between September and November 2020, 11 interviews ranging from 23 to 55 min were conducted physically in each nursing home, with a mean duration of 42 min. Participant characteristics are presented in Table 4.

Data from the interviews produced five thought clusters related to the critical factors that were considered when implementing a telemedicine project. As shown in Table 5, these clusters were assembled to generate the following three major themes: (1) motivations of nursing homes to use telemedicine, (2) building a relevant telemedicine medical offer, and (3) organizing telemedicine in a nursing home.

#### 3.2.1. Motivations of Nursing Homes to Use Telemedicine

The use of telemedicine by nursing homes is guided by two types of motivation, namely improving access to care for their residents, and improving cooperation between healthcare professionals.

For rural nursing homes located far from the general hospital, transportation increases the burden of consultations for patients, so they sometimes forego organizing face-to-face consultations for their residents. In this context, interviewees considered telemedicine was highly beneficial since it reduces the spread between time for transportation and effective consultation.

“*[Our expectations] were to avoid transportation for residents who do not necessarily need to go to the hospital in-person and to avoid waiting times once there. Spending four hours on a temporary bed for only 15 min of medical consultation is something that our residents do not accept anymore.*” (Rur_1_Nurse)

For urban nursing homes, telemedicine was considered more as a method to provide quicker specialized medical consultations or opinions compared to face-to-face consultations.

“*I would say [that our expectations] essentially concerned prevention; to be able to see specialists quickly without having to bring older patients to the hospital with all the related consequences.*” (Urb_6_Dir)

Telemedicine was also considered to improve cooperation between nursing homes and the physicians involved in patient management. A direct teleconsultation between a nursing home and a physician allows the latter to propose better treatment options and therapeutic strategies for patients.

“*When [the residents] go to an in-person consultation, we cannot be there with them. Indeed, the doctor can encounter some difficulties, or may suggest treatment strategies that we don’t necessarily understand.*” (Rur_1_Nurse)

#### 3.2.2. Building a Relevant Telemedicine Medical Offer

The telemedicine medical offer should be developed based on patients’ needs, and more importantly on the inability of treating physicians to address them. Treating physicians are conductors of the coordinated patient pathways, and a telemedicine medical offer must consider areas where they believe that they cannot manage the situation. In the present study, a telemedicine offer with diabetologists from the nursing home’s point of view does not fit their needs because treating physicians are comfortable with managing non-insulin-required diabetes.

“*To present the telemedicine project to us, the [telemedicine project manager] from the hospital used the example of diabetes. I do not think this was a good choice. General practitioners know how to manage diabetes, particularly for 90-year-old patients.*” (Rur_3_Nurse)

Even if connecting nursing homes and the hospital is necessary (specifically when creating a fast-track care pathway), telemedicine projects in nursing homes should focus on facilitating relations with the usual physicians surrounding a patient.

“*We needed a solution that is interoperable with the hospital because it is the reference center. However, our electronic health records included a telemedicine feature which would have allowed us, via a simple link, to provide teleconsultations with the known ambulatory care treating physicians and specialists, especially in cases where they cannot come to the nursing home in a reasonable time.*” (Urb_8_Dir)

#### 3.2.3. Organizing Telemedicine in Nursing Home for Time and Space

Telemedicine management requires a reorganization of time as well as the necessary workspace at the nursing home, whereas a physical consultation only requires the nursing home team to prepare the patient for transportation. A teleconsultation requires more time from the staff during the entire consultation and requires them to remain with the patient to fill out medical forms. This can only be managed if a new workplace organization is implemented to avoid heavy workloads.

“*An external appointment requires scheduling the appointment and the transportation; however, we are free during the consultation. For a teleconsultation, there is a lot of preparation plus the time spent in the teleconsultation. If we had to do more teleconsultations, we would not be able to adequately manage due to the lack of nursing resources.*” (Urb_8_Dir)

In our study, the nursing homes only used teleconsultations; however, the participants stated that some situations would be more efficiently managed by employing asynchronous tele-expertise to obtain a specialist opinion without too much impact on their work time.

“*For wound care, I know the physician well, and I have her phone number. When there is a pressure ulcer, I send her a picture and she replies back to me by text. We do not use synchronous teleconsultations because it is too difficult to organize. A picture is much quicker!*” (Rur_2_Doc)

Two types of workspace management have been used in this project: the creation of a dedicated room for teleconsultations and the use of a mobile cart. A dedicated room allows for a guarantee of patient confidentiality and provides the necessary bandwidth required for this type of communication. Unfortunately, it is not adapted to bedridden and/or dependent patients. The use of mobile carts or tablets seemed more effective in conditions where a sufficient Wi-Fi network was present.

“*We started to do teleconsultations using the nursing home Wi-Fi, but we quickly changed our strategy. Consultations were performed at the nursing stations or in the resident’s room. Bandwidth was not the same everywhere. We have been using a 4G key since then. Everything works very well.*” (Rur_4_Nurse)

## 4. Discussion

We conducted a mixed-method study to describe the telemedicine activity of nursing homes cooperating with a general hospital while identifying the critical factors to consider when implementing a telemedicine project.

Our quantitative study showed that there were different needs in terms of frequency and the types of medical specialties requested depending on whether a nursing home was urban or rural. The consumption of general practice telemedicine for rural nursing homes was sixteen-fold higher compared to urban ones, which matched with literature. Rural zones suffer from a greater lack of general practitioners than urban zones [39], but general practitioners are the cornerstone of the care pathway of elderly patients who have several comorbidities, including residents of nursing homes [40].

Furthermore, our qualitative study revealed that three critical factors are required for the implementation of a telemedicine project in nursing homes: (1) motivations of nursing homes to use telemedicine, (2) building a relevant telemedicine medical offer, and (3) finding specific organization.

In our study, nursing home staff confessed that they may forego face-to-face consultations because of the associated burden to a patient during transportation. This result matched with Parsons et al. which was a systematic review highlighting that the long-distance for accessing specialized services for rural older adults leads to delayed care [41]. These results are in line with literature on ageism (otherwise known as age discrimination), which shows a strong association between age criterion and the decision whether to provide a treatment or intervention [42], and telemedicine in nursing homes may be a way to counter ageism.

The development of chronic diseases among the elderly population is leading to increasingly complex care pathways requiring round-trips between city care and hospitals [43,44]. Telemedicine can be a tool to link these different components of care and reinforce cooperation between healthcare professionals [45,46]. However, our results show that it is necessary to pay attention to the care offers to be built in connection with the attending physician. Moreover, it is important to privilege the relationship of patients with their usual practitioners.

Our study shows that a teleconsultation (synchronous video consultation) is considered particularly time-consuming by nursing homes’ teams, and for some needs, tele-expertise may be more relevant and should be promoted. Tele-expertise is a type of asynchronous telemedicine practice for health professionals to request an expert medical opinion through a dedicated software or secured e-mail system [47]. According to the results of the eConsult study, tele-expertise allows professionals to access a high level of expertise that is difficult to access in long-term care facilities while limiting the impact on participants’ time [21]. However, to make tele-expertise sustainable, it is necessary to think about integrating it into existing workflows to not transfer any potential burdens from the requesting healthcare professional to the required experts [48].

As telehealth becomes ubiquitous, it is critical to consider its potential to exacerbate disparities in care, especially considering accessing technology and digital literacy [49]. Nursing homes have had to organize their teleconsultation services according to their broadband access, especially when they are rural as shown in our findings. This is in line with Cortelyou-Ward et al., which identified that rural community members face a double-burden: the lack of physicians and the lack of high-quality broadband access [50]. In this context, increasing broadband access is considered a social determinant of health.

Consequently, each telemedicine project needs to be carefully defined to meet the needs of the nursing home. Different needs may be present for nursing homes that are in the same territory even if they interact with the same hospital. It is therefore required to involve all relevant stakeholders in the project definition. Having a dedicated telemedicine project manager within the nursing home may be an effective way to increase the chances of success even if we were unable to exactly assess this specific factor within this study [51].

The limits of this study included the lack of integration of non-verbal information in the analysis of the interview, the single point-of-view from the nursing home representatives which could have benefited from additional interviews with the hospital physicians as well as the residents, and the potential lack of universality of the results due to the specificity of the French healthcare model. The limited volume of teleconsultations did not allow us to conduct more advanced statistical modeling that may have shown more detailed results, for example, by adjusting the results on medical density as a potential confusion factor.

Even though the deployment of telemedicine projects accelerated in the recent pandemic, we did not observe any effects on the use of telemedicine in our quantitative study from COVID-19. Moreover, all our interviews were conducted after the beginning of the pandemic between the first two lockdowns in France.

Even though patients were specifically asked the question (In case of a new public health crisis, what would you expect from a telemedicine solution?), the interviewees’ answers were not specifically related to COVID-19, and this did not allow us to identify it as an additional critical factor.

## 5. Conclusions

This mixed-method study showed the various behaviors of nursing homes regarding video teleconsultations activities with a general hospital related to their urban or rural setting. Quantitative analysis showed the higher use of general practice support for rural nursing homes. Qualitative analysis identified three factors required when defining a telemedicine project in nursing homes: motivation to perform teleconsultations, the telemedicine offer, and the specific organization that is needed to build. To maximize the success of these projects, various elements presented in this study should be taken into account.

## Figures and Tables

**Table 1 ijerph-18-11148-t001:** Distribution of the teleconsultations during the three-year period according to the type of nursing homes (rural or urban).

Type of NH	Apr 2018 to Apr 2019 *n* (%)	Apr 2019 to Apr 2020 *n* (%)	Apr 2020 to Apr 2021 *n* (%)
Rural NH	79 (77%)	170 (78%)	214 (80%)
Urban NH	23 (23%)	49 (22%)	55 (20%)
Total NH	102 (100%)	219 (100%)	269 (100%)

NH: Nursing home.

**Table 2 ijerph-18-11148-t002:** Characteristics according to localization (rural or urban) performed between nursing homes and Troyes Hospital in France between 2018 and 2021.

Specialty	Rural NH, *n* (%)	Urban NH, *n* (%)	Total NH, *n* (%)
Dermatology	20 (4%)	18 (14%)	38 (6%)
Diabetology	44 (10%)	18 (14%)	62 (11%)
Genetics	2 (0%)	0 (0%)	2 (0%)
Geriatrics	35 (8%)	12 (9%)	47 (8%)
General Practice	254 (55%)	11 (9%)	265 (45%)
Wound healing	105 (23%)	67 (53%)	172 (29%)
Pneumology	1 (0%)	1 (1%)	2 (0%)
Rheumatology	1 (0%)	0 (0%)	1 (0%)
Urology	1 (0%)	0 (0%)	1 (0%)
Total	463 (100%)	127 (100%)	590 (100%)

NH: Nursing home.

**Table 3 ijerph-18-11148-t003:** Multivariate analysis with a negative binomial model evaluating the number of teleconsultations during aggregated 1-year periods with Troyes Hospital in France between 2018 and 2021.

Overall	Adjusted on Time				Fully Adjusted			
Variable	RR	95% CI Low	95% CI High	*p* Value	RR	95% CI Low	95% CI High	*p* Value
**All specialties ^a^**								
Type: rural (reference = urban)	2.659	1.205	5.713	0.016	2.484	1.083	5.518	0.03
Number of residents (continuous variable)	0.992	0.981	1.006	0.23	0.996	0.985	1.010	0.57
**General practice ^b^**								
Type: rural (reference = urban)	15.062	3.099	69.417	<0.001	16.305	3.505	73.523	0.001
Number of residents (continuous variable)	0.967	0.933	0.999	0.002	0.965	0.933	0.995	0.02

^a^: Null deviance: 90.258 on 77 degrees of freedom; Fully adjusted model residual deviance: 83.975 on 73 degrees of freedom. ^b^: Null deviance: 63.473 on 77 degrees of freedom; Fully adjusted model residual deviance: 44.530 on 73 degrees of freedom.

**Table 4 ijerph-18-11148-t004:** Characteristics of the telemedicine project managers interviewed for the study about nursing homes and Troyes hospital in France between 2018 and 2021.

ID	Telemedicine Project Manager	Urban or Rural
Rur_1_Nurse	Coordinating nurse	Rural
Rur_2_Doc	Coordinating physician	Rural
Rur_3_Nurse	Coordinating nurse	Rural
Rur_4_Nurse	Coordinating nurse	Rural
Rur_5_Doc	Coordinating physician	Rural
Rur_6_Doc	Coordinating physician	Rural
Urb_7_Dir	Nursing home director	Urban
Urb_8_Dir	Nursing home director	Urban
Urb_9_Nurse	Coordinating nurse	Urban
Urb_10_Dir	Nursing home director	Urban
Urb_11_Doc	Coordinating physician	Urban

**Table 5 ijerph-18-11148-t005:** Summary of the overall themes derived from the interviews and their clustering of nursing home using telemedicine.

Major Themes	Clusters	Subclusters
Motivations of nursing homes to use telemedicine	Why use telemedicine in nursing homes?	Improving access to care
		Improving cooperation between health professionals
Building a relevant telemedicine medical offer	What to do with telemedicine?	Medical offer based on the patients’ needs
		Medical offer based on the treating physicians’ needs
	Who to do telemedicine with?	Hospital’s physicians
		Usual patients’ physicians
Organizing telemedicine in nursing home	Time considerations	Teleconsultation is time consuming for the nursing home staff
		Asynchronous tele-expertise?
	Spatial considerations	Dedicated room for telemedicine
		Telemedicine provided at the patients’ bedside

## Data Availability

The data presented in this study are available upon the request to the corresponding author. The data from this study are not accessible to the public due to French legislation which makes the investigators responsible for data processing.

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
