# Peer review of "The Use of Telemedicine in Nursing Homes: A Mixed-Method Study to Identify Critical Factors When Connecting with a General Hospital"

_ijerph, 2021, doi:10.3390/ijerph182111148_

Round 1

Reviewer 1 Report

The authors studied the telemedicine activity in nursing homes when cooperating with a general hospital. They conducted a mixed-method study using quantitative and qualitative methods. Their qualitative study showed that four critical factors are required for the implementation of a telemedicine project in nursing homes: motivation to perform teleconsultations, the medical needs of the nursing homes, healthcare professionals willing to engage in the project, and specific organization. It is an interesting work. I have several questions:

  1. Are there similar researches conducted in other countries? Can the authors do some comparison studies?
  2. The manuscript needs to be carefully checked. For example, “Data Availability Statement: In this section, please provide details regarding where data supporting reported results can be found, including links to publicly archived datasets analyzed or generated during the study. Please refer to suggested Data Availability Statements in section “MDPI Research Data Policies” at https://www.mdpi.com/ethics. You might choose to exclude this statement if the study did not report any data” should be replaced with your own words. You can’t put the example sentences in your manuscript.
  3. There were many sentences that seemed to be direction quote, such as “[Our expectations] were to avoid transportation for residents who do not necessarily need to go to the hospital in-person and to avoid waiting times once there. Spending four hours on a temporary bed for only 15 minutes of medical consultation is something that our residents do not accept anymore.” (Rur_1_Nurse). The authors need to summary the key points and represent with figures and tables, rather than directly quote the sentences.
  4. In Results, there were no references at all. Please cite previous researches to support your results.
  5. If possible, the authors should provide the data as supplementary materials.

Author Response

Thank you very much for your feedback on our work. Please find enclosed our responses to your comments.

Reviewer 2 Report

The research investigated the use and impact of telemedicine in nursing homes. Data came from three years of twenty-six nursing homes conducting teleconsultations with a general hospital in France, from April 2018 to April 2021. The research reported four critical factors required for the implementation of a telemedicine project in nursing homes, including motivation to perform teleconsultations, the medical needs of the nursing homes, healthcare professionals willing to engage in the project, and specific organization. It also showed different uses of teleconsultations between the rural or urban localization of nursing homes. The research helps in a better understanding of the implementation and quality improvements of telemedicine.

The paper, however, seems to be unclear along several lines making the introduction and analysis aspects appear weak. Major revision is needed in several aspects.

  1. “To understand the motivations of connecting with a general hospital” does not seem to be the major research aim or the major result of the research.
  2. The research aim, results and conclusion did not match very well. It is unclear whether the paper focused on the activity, motivation, or impact of urban and rural nursing homes of using telemedicine to connect with a general hospital. Also, the result part and discussion part can be better organized to stress the key findings better.
  3. Introduction part. The introduction provides very limited information on previous research, research gaps, and the significance of this paper.
  4. Method part & result part.

Why did the Poisson models only account for time (aggregated 1-year periods), location (urban or rural), and the number of residents in the nursing homes? Are these the most significant risk factors based on previous research? How good was the final model?

Since COVID-19 brings significant impacts to the use of telemedicine, it is important to separate and report the situations before and during the COVID-19 pandemic.

For most of the interview questions, the paper just provided one answer citation to each question. It is important but unclear how consistent or different across the different respondents.

  1. The major findings should be better addressed and compared with related research.
  2. Tables and figures. Figure 1 doesn’t provide a lot of helpful information. A brief paragraph on the locations, population, characteristics, and distance of the nursing homes from the hospital could be better.

Author Response

Thank you very much for your feedback on our work. Please find our responses to your comments.

Reviewer 3 Report

This is a well written and clearly presented paper. The study would have been significantly enhanced if the patients views of the consultations was sought and represented.......

I feel it only needs minor revision......

I would suggest the authors review the Discussion section, specifically line 224 to 231. The issue of provision of specialty care needs to be expanded on further so the reader can fully understand the reasoning behind the comments. Also the use of asynchronous consultation should be expanded on so readers understand what it means and why it could add real value.

Author Response

(The authors gave the same response as above.)

Reviewer 4 Report

The présent study amis at using mixed-methods design to assess the extent of the use of telemedicine approches in nursing in the COVID-19 context. The study tends to figure out the difference between the use of telemedicine in urban and rural areas.

Instead of mixed-method, the abstract reports the results of the quantitative study.

The method session is limited to show how both qualitative and quantitative analyses are complementary for precise insight in the study outcome.

I also miss the results of the qualitative research.

There authors are advised to revise and clearly choose the appreciate study design.

The conclusion is biased and need to be revised.

Author Response

(The authors gave the same response as above.)

Round 2

Reviewer 2 Report

In general, the revision responds to all my comments and advice well. However, the abstract could be improved, especially the background and aim. Also, there are some typos and errors. A thorough check would be better.

Author Response

Thank you very much for having reviewed our work. 
Please find enclosed our answers to your comments. 

Reviewer 4 Report

Thanks for the revised version of your paper. This version clearly shows why you use the mixed-methods MM) and how. Though you forgot to say which type of MM you 1) concurrent or 2) sequential - explanatory or explorative?

The abstract should reflect the why and how of the MM and summarize the conclusion.

The big weakness of the present study is the lack of an appropriate literature review to support your theory and beyond this to show the novelty of this work.

Advice: Could you please add a section for related work and show that this work is covering a gap in the literature? This will add more value to your work. The one provided in the introduction is too short.

Author Response

(The authors gave the same response as above.)
